



# Impact of a Two-Dimensional Steep Hill on Wind Turbine Noise Propagation

Jules Colas[1], Ariane Emmanuelli[1], Didier Dragna[1], Philippe Blanc-Benon[2], Benjamin Cotté[3], and Richard Stevens[4]

[1]Ecole Centrale de Lyon, CNRS, Universite Claude Bernard Lyon 1, INSA Lyon, LMFA, UMR5509, 69130, Ecully, France
[2]CNRS, Ecole Centrale de Lyon, INSA Lyon, Universite Claude Bernard Lyon 1, LMFA, UMR5509, 69130, Ecully, France
[3]Institute of Mechanical Sciences and Industrial Applications (IMSIA), ENSTA Paris, CNRS, CEA, EDF, Institut Polytechnique de Paris
[4]Physics of Fluids Group, Max Planck Center Twente for Complex Fluid Dynamics, ,
J. M. Burgers Center for Fluid Dynamics, University of Twente, P. O. Box 217, 7500 AE Enschede, The Netherlands

**Corresponding author:** Jules Colas, jules.colas@ec-lyon.fr

**Abstract.** Wind turbine noise propagation in a hilly terrain is studied through numerical simulation in different scenarios. The linearized Euler equations are solved in a moving frame that follows the wavefront, and wind turbine noise is modeled with an extended moving source. We employ large eddy simulations to simulate the flow around the hill and the wind turbine. The sound pressure levels obtained for a wind turbine in front of a 2D hill and a wind turbine on a hilltop are compared to a baseline flat case. First, the source height and wind speed strongly influence sound propagation downwind. We find that topography influences the wake shape inducing changes in the sound propagation that drastically modify the SPL downwind. Placing the turbine on the hilltop increases the average sound pressure level and amplitude modulation downwind. For the wind turbine placed upstream of a hill, a strong shielding effect is observed. But, because of the refraction by the wind gradient, levels are comparable with the baseline flat case just after the hill. Thus, considering how terrain topography alters the flow and wind turbine wake is essential to accurately predict wind turbine noise propagation.

## 1 Introduction

The increase in demand for renewable energy and the low power density characteristic of wind energy has led to the development of extended wind farms. A significant barrier to the successful implementation of these installations is noise annoyance, which often leads to diminished public acceptance (Gaßner et al., 2022). Hence, accurate models for noise prediction from single wind turbines and wind farms are necessary to extend the use of wind energy. In recent years, significant efforts have been made to characterize wind turbine noise. This includes the study of the wind turbine inflow, noise emission mechanisms, and the effect of the propagation medium on the far-field noise level. Aerodynamic noise, which is caused by the interaction of the incoming wind with the blades of the turbines, is dominant, broadband and can propagate over several kilometers (Van Den Berg, 2004; Hansen et al., 2019). The main origins of aerodynamic noise are turbulent inflow noise and trailing edge





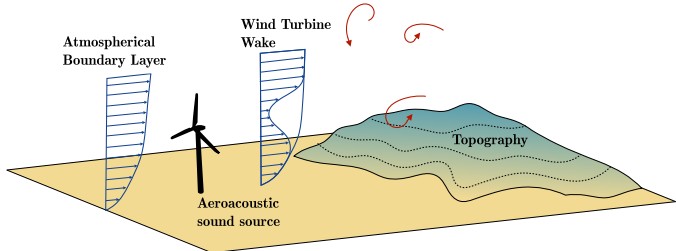

**Figure 1.** Sketch of the main wind turbine noise propagation factors.

noise (Oerlemans et al., 2007). The movement of the blades induces amplitude modulation (AM), which is considered to be one of the main annoyance issues (van den Berg, 2005; Hansen et al., 2019). The noise production is greatly influenced by the geometry of the blades, and by the incoming wind profile and turbulence characteristics. Several models have been developed

to predict wind turbine noise emission. Point source approaches are commonly utilized (Lee et al., 2016; Prospathopoulos and Voutsinas, 2007). However, more sophisticated methods considering an extended aeroacoustic sound source have also been developed (Cotté, 2019; Bresciani et al., 2023; Cao et al., 2020). These approaches enable more accurate modeling of the flow and the geometry's impact on sound levels and AM prediction. Atmospheric conditions affect wind turbine noise propagation, prompting the development of various numerical methods to study these effects. They are generally based on propagating

sound through already generated flow fields, either analytically or numerically. Predictions of wind turbine noise propagation using a parabolic equation model or particle-based approach have been compared against field experiments (Könecke et al., 2023; Nyborg et al., 2023). Barlas et al. (2018) and Heimann and Englberger (2018) have studied the influence of wind velocity and temperature profiles on noise generation mechanisms and noise propagation. An important finding is that the wind turbine wake can act as a waveguide and create a focusing zone near the ground. Its position and intensity notably depend on the

velocity deficit of the wind turbine wake. One of the main results (Barlas et al., 2017b; Heimann and Englberger, 2018) is that far-field AM is greatly influenced by wind shear and turbulent intensity and that focusing zones (their existence, intensity and position) strongly depend on small variations in the meteorological conditions.

Besides atmospheric refraction, factors such as topography and ground absorption also affect long-range sound propagation, as demonstrated in Fig. 1. Placing wind turbines on hilltops is common due to the improvement in energy production (Berg

et al., 2011). Therefore, in such configurations, considering the topographical effects on noise production and propagation becomes essential (Elsen and Schady, 2021). The influence of a 3D hill on both the flow around the wind turbine and noise propagation has been studied by Heimann et al. (2018). In their work, the atmospheric flow perturbed by the hill and the wind turbine is computed using Large Eddy Simulation (LES). The results are then used as an input for propagation simulations using a 3D ray-based sound particle model. The authors show that positioning the wind turbine on the hilltop can reduce the overall

sound pressure level (OASPL) in the far field. This can compensate for the amplification on the ground due to the wake, which tends to enhance the OASPL further downstream. Shen et al. (2019) studied topography effects for a realistic case considering a wind farm over a complex ground. They used a 2D parabolic equation method to propagate sound over long distances and



account for topography and flow effects (Sack and West, 1995; Nyborg et al., 2023). The noise levels were compared with those for a flat ground considering either a homogeneous atmosphere at rest or a logarithmic mean velocity profile. Both the topography and the turbulent flow characteristics were shown to be key components in predicting noise propagation. Despite these studies, the combined effect of the flow and topography on sound propagation is not yet fully understood. Indeed, noise levels in the far field are strongly dependent on parameters like the position of the sources, hillshape, and atmospheric boundary layer (ABL) conditions, which makes it difficult to draw a general conclusion on the effect of topography. Moreover, the impact of the turbine's position relative to the hill is largely unexplored, and the effects of atmospheric flow conditions on the interplay between sound production and propagation remain unclear.

This work aims to address these last two points and further investigate the phenomena present when considering wind turbine sound propagation with topography. Hence sound propagation is simulated for several positions of a single wind turbine relative to a 2D hill, i.e a ridge. The position of the wind turbine relative to the hill is expected to have an impact both on the flow around the wind turbine, especially the development of the wake, and directly on sound propagation, through reflection, refraction and scattering of acoustic waves. To study these phenomena, a propagation method based on the linearized Euler equations (LEE) (Colas et al., 2023) is used. The effects of the hill and the wind turbine on the flow are taken into account through the mean flow values obtained from LES performed in the work of Liu and Stevens (2020). LEE methods are not widely used for the long-range propagation of wind turbine noise because they are computationally demanding. Here the use of LEE is relevant as more classical methods based on geometrical acoustics or parabolic equations could be limited in the presence of topography.

The paper is organized as follows. In Sec. 2, the methodology is briefly described. Then the cases studied and the numerical simulations performed are presented in Secs. 3 and 4. The propagation effects are first described in Sec. 5.1. Then the SPL fields for the complete wind turbine are compared between the different cases in Sec. 5.2. Finally, we compare our results with previous studies in Sec.6 and give concluding remarks in Sec. 7.

## 2   Methodology

### 2.1   Wind turbine noise model

The methodology employed to compute wind turbine noise involves three steps as depicted in Fig 2. It is similar to the methodology described in Colas et al. (2023) and is only briefly summarized here. First, LES with an immersed boundary method (Gadde et al., 2021; Gadde and Stevens, 2019; Stieren et al., 2021; Stevens et al., 2018) is used to determine the average flow field. Although the simulations are unsteady, only the mean velocity fields are utilized in this study, disregarding turbulence scattering despite its known influence on wind turbine noise propagation.

The wind turbine noise is computed using the moving monopole (MM) approach proposed by Cotté (2019). First, the SPL in the free field, denoted $\mathrm{SPL_{ff}}$, is computed using the Amiet strip theory (Amiet, 1976) Each blade is divided into several segments, considered as uncorrelated point sources. Then, the trailing edge noise and turbulent inflow noise are calculated considering the blade geometry and input mean flow, for each segment at each angular position (Tian and Cotté, 2016; Mascarenhas et al., 2022). Note that this model is only valid in the far field (distances larger than the blade chord and the wavelength). The




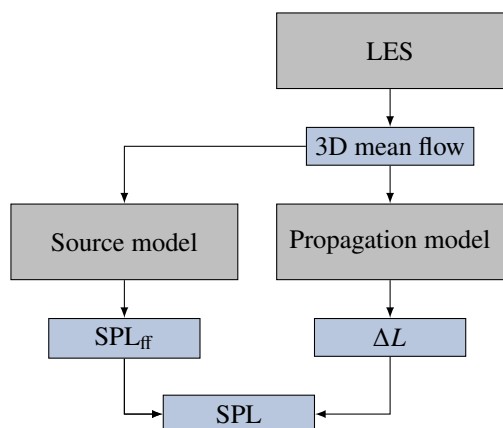

**Figure 2.** Diagram of the complete wind turbine noise prediction methodology.

SPL at a receiver is finally computed by adding the effect of atmospheric absorption and the relative sound pressure level, denoted $\Delta L$, which encompasses the ground and refraction effects on the sound propagation. It writes:

$$\mathrm{SPL}^i(\mathbf{x}, f, \beta) = \mathrm{SPL}_{\mathrm{ff}}^i(\mathbf{x}, f, \beta) + \Delta L^i(\mathbf{x}, f, \beta) - \alpha(f)R, \tag{1}$$

where $i$ is the segment index, $\mathbf{x}$ the receiver coordinates, $f$ the frequency, $\beta$ the blade angle (with $\beta = 0$ coresponding to
85 the blade pointing upwards, see Fig 7), $\alpha$ the atmospheric absorption coefficient, and $R$ the distance from source to receiver. In this case, $\Delta L$ is computed using 2D LEE simulations, described in the next section, but other propagation models could be employed like parabolic equation methods, ray tracing (see Sec. 5.1.3) or engineering models. To decrease computational expenses, $\Delta L$ is not calculated for every segment position. Instead, fictive source heights are assumed along a vertical line in the rotor plane that passes through the turbine's hub. Hence $\Delta L^i$ in Eq. (1) corresponds to a linear interpolation of the
90 computed values of $\Delta L$ between the two closest fictive sources. Finally, the SPL produced by the wind turbine is obtained by summing the contributions of all blade segments of the three blades:

$$\mathrm{SPL}(\mathbf{x}, f, \beta) = 10 \log_{10} \left( \sum_{i=1}^{3N_s} 10^{\mathrm{SPL}^i(\mathbf{x}, f, \beta)/10} \right), \tag{2}$$

where $N_s$ is the number of segments per blade. The SPL are integrated for a set of frequencies from 50 Hz to 1 kHz (Colas et al., 2023) to retrieve the OASPL:

$$95 \quad \mathrm{OASPL}(\mathbf{x}, \beta) = 10 \log_{10} \left( \sum_{i=1}^{N_f} \Delta f_i 10^{\mathrm{SPL}(\mathbf{x}, f_i, \beta)/10} \right), \tag{3}$$

where $N_f$ is the number of computed frequencies and $\Delta f_i$ is the band width. Finally, the averaged OASPL ($\overline{\mathrm{OASPL}}$) and AM are computed over one rotation such that:

$$\overline{\mathrm{OASPL}}(\mathbf{x}) = 10 \log_{10} \left( \sum 10^{\mathrm{OASPL}(\mathbf{x}, \beta)/10} / N_\beta \right),$$

$$\mathrm{AM}(\mathbf{x}) = \max_\beta(\mathrm{OASPL}(\mathbf{x}, \beta)) - \min_\beta(\mathrm{OASPL}(\mathbf{x}, \beta)), \tag{4}$$



where $N_\beta$ is the number of angles discretizing the rotor rotation.

## 2.2 Propagation model

The propagation model is based on the numerical solution of the LEE, which are solved in a 2D curvilinear mesh to account for topography. A set of two equations is derived from the LEE for atmospheric acoustics without source terms (Ostashev et al., 2005):

$$
\begin{aligned}
&\frac{\partial p}{\partial t} + \mathbf{V_0}.\nabla p + \rho_0 c_0^2 \nabla \cdot \mathbf{v} = 0, \\
&\frac{\partial \mathbf{v}}{\partial t} + (\mathbf{V_0}.\nabla)\mathbf{v} + (\mathbf{v}.\nabla)\mathbf{V_0} + \frac{\nabla p}{\rho_0} = 0,
\end{aligned}
\tag{5}
$$

with $p$ and $\mathbf{v} = (u,w)$ the acoustic pressure and velocity, $\rho_0$ the mean density, $\mathbf{V_0} = (u_0, w_0)$ the mean velocity, and $c_0$ the sound speed. A conservative formulation of this system of equations is then derived for the transformation of the coordinate system from Cartesian to curvilinear. The same curvilinear transformation as in Colas et al. (2023) and originally proposed by Gal-Chen and Somerville (1975) is employed to follow the terrain elevation. A Gaussian pulse introduced as an initial condition is used to represent a broadband monopole (Colas et al., 2023). Eqs. 5 are solved with high-order optimized finite-difference techniques (Bogey and Bailly, 2004; Berland et al., 2006). To simulate a realistic ground, a broadband impedance condition (Troian et al., 2017) is implemented at the bottom of the domain and a convolutional perfectly matched layer (CPML) simulates unbounded propagation at the top of the domain (Cosnefroy, 2019; Petropoulos, 2000). The acoustic variables and the boundary conditions are computed in a moving frame that follows the wavefront. This greatly reduces the computational cost as the propagation distance is large (around 15000 acoustic wavelengths). This method is described in Emmanuelli et al. (2021) and Colas et al. (2023).

From the time domain solution, it is possible to recover a frequency domain solution. The $\Delta L$ can be derived from the time signal $p(t)$ recorded at one receiver such that:

$$
\Delta L(f, \mathbf{x}) = 10 \log_{10}\left( \frac{|P(f,\mathbf{x})|^2}{|P_{\text{ff}}(f,\mathbf{x})|^2} \right),
\tag{6}
$$

with $P$ the Fourier transform of $p$ and $P_{\text{ff}}$ the free field solution, i.e. the solution for the same source term but without any mean flow or ground reflection. The $\Delta L$ for each third-octave band $f_c$ is computed by averaging over the broadband results.

## 3 Cases studied

This work focuses on three configurations that were previously studied by Liu and Stevens (2020). For each configuration, LES are performed for a truly neutral ABL, with and without the turbine inside the flow. These cases are referred to as $X_{\text{WT}}$ (with the wind turbine) and $X_{\text{ABL}}$ (without the wind turbine). This allows one to investigate topographic effects on sound propagation isolated from wake effects. The configurations are defined as such:





   – Case $A_{ABL}$ and $A_{WT}$: baseline case of a wind turbine with flat ground

   – Case $B_{ABL}$ and $B_{WT}$: a wind turbine placed upstream of a 2D hill

   – Case $C_{ABL}$ and $C_{WT}$: a wind turbine placed on top of a 2D hill

The hill considered for cases B and C is defined such that

$$h(x) = h_{\max}\cos^2\left(\frac{\pi x}{2l}\right), \quad -l \leq x \leq l \tag{7}$$

where $h_{\max} = 100$ m and $l = 260$ m. Note that $x = 0$ corresponds to the turbine position. Hence the hill is shifted between case C and case B (see Fig. 5); $x = 0$ corresponds to the beginning of the hill in case B and to the hilltop in case C. Two wind profiles are considered such that the wind speed at $z = 100$ m is equal to $u_{\mathrm{ref}} = 8$ m.s$^{-1}$ and $u_{\mathrm{ref}} = 10$ m.s$^{-1}$, which are typical values for wind turbine applications. The diameter and height of the turbine are set to 100 m and the roughness height at the ground is 0.01 m, which is representative for flow over smooth terrain. As we consider neutral conditions, the temperature is assumed constant throughout the domain. The variable porosity model is used to model the ground impedance (Attenborough et al., 2011). The effective flow resistivity is set to $50$ kNs.m$^{-4}$ and the effective porosity change rate to $100$ m$^{-1}$, which reflect natural soil conditions (Cotté, 2018).

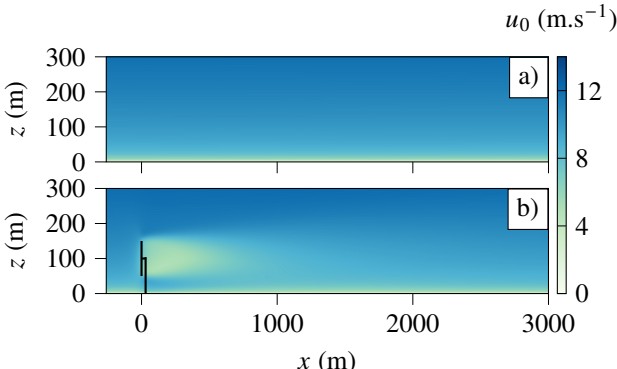

**Figure 3.** LES mean flow velocity fields for the baseline case a) $A_{ABL}$ without and b) $A_{WT}$ with the wind turbine. See Fig. 4 for $yz$ plane view.

    The side view of the streamwise mean velocity is plotted in the plane of the wind turbine's hub in Fig. 3 for cases $A_{ABL}$ and $A_{WT}$ (note that for conciseness only the results for $u_{\mathrm{ref}} = 10$ m.s$^{-1}$ are presented in this section). When a wind turbine is present, the velocity deficit that occurs downstream requires several hundred meters to recover. The shape of the wake can also be observed in the $yz$ plane at different downstream positions in Fig. 4. It presents a circular zone with a few meters per second velocity deficit just after the wind turbine, that progressively fades away further downstream of the wind turbine.

    In the presence of the hill, the flow is first shown without the wind turbine in Fig. 5a. An increase in speed is observed at the hilltop as the flow accelerates around it. A recirculation zone is then created downstream of the hill, which leads to the

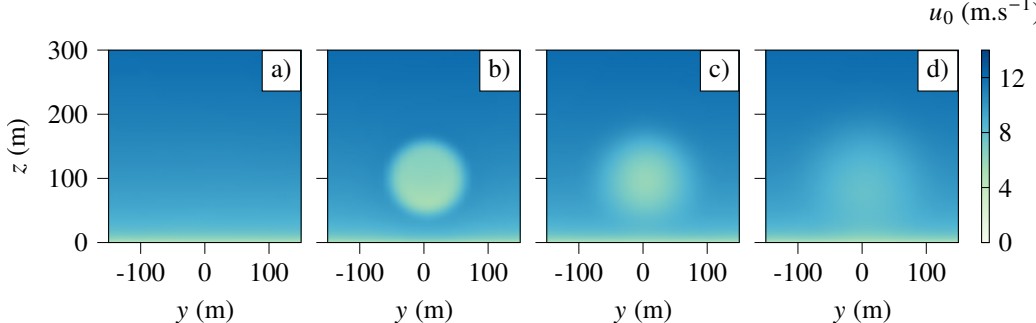

**Figure 4.** Case $A_{WT}$: mean flow from LES in the $yz$ plane at positions a) $x = -100$ m, b) $x = 0$ m, c) $x = 100$ m, and d) $x = 200$ m, with $x = 0$ m the turbine's location. See Fig. 3 for $xz$ plane view.

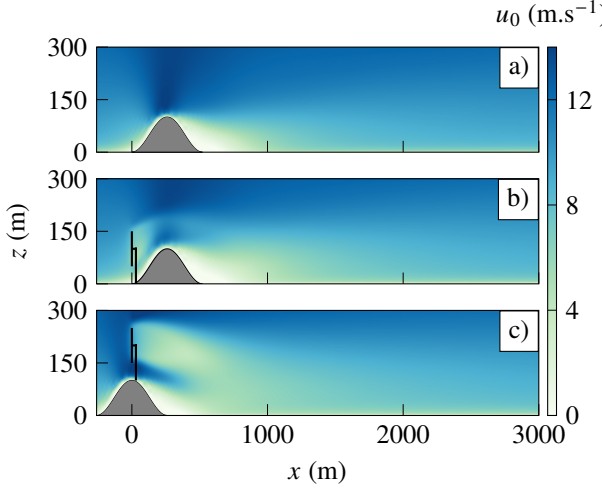

**Figure 5.** Mean flow velocity field for cases a) $B_{ABL}$, b) $B_{WT}$, and c) $C_{ABL}$.

reduction of the mean velocity towards negative values. With the wind turbine in front of the hill (Fig. 5b) the wake of the turbine follows the shape of the hill, counterbalancing the increase of windspeed at the hilltop. Hence the hill's wake appears longer and higher. With the wind turbine on the hilltop (Fig. 5c), its wake is larger and mixes with the hill's wake.

Profiles of the streamwise mean velocity are plotted for cases $A_{WT}$, $B_{WT}$ and $C_{WT}$ in Fig. 6 at different distances from
150 the wind turbine to better show the effect of the hill. For all cases, the velocity deficit has a top-hat shape for the first few hundred meters downwind of the turbine due to the actuator disc model (Stevens and Meneveau, 2017; Bastankhah and Porté-Agel, 2014). Then, as the ABL flow recovers, a Gaussian shape seems more accurate to describe the wake. Another interesting observation is the influence of the topography on the wake shape. For case $B_{WT}$ (Fig. 6b) the wake goes up and then down to follow the shape of the hill. The wind speed gradients at the top and bottom of the wake are smaller than for the baseline
case $A_{WT}$ (Fig. 6 a). For case $C_{WT}$ (Fig. 6c) the turbine's wake moves down to follow the topography. The wind gradients

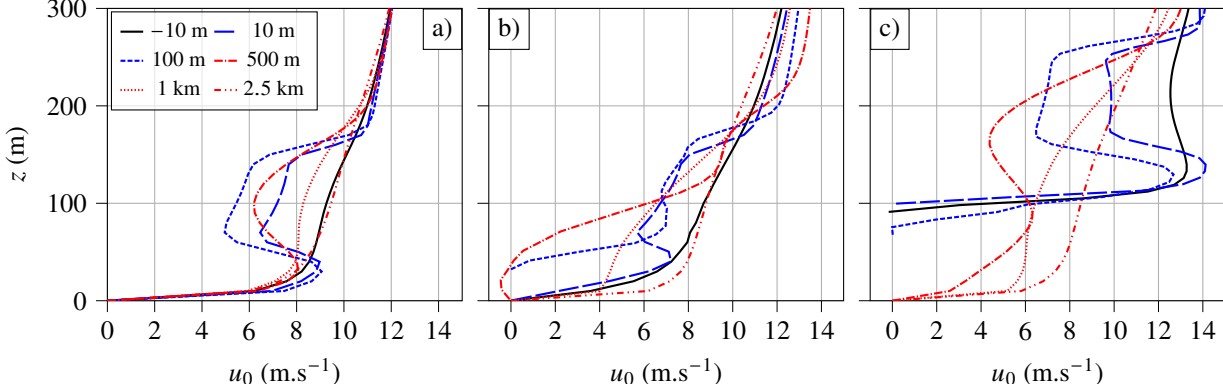

**Figure 6.** Mean flow velocity profiles at different positions $x$ from -10 m to 2.5 km for cases a) $A_{WT}$, b) $B_{WT}$, and c) $C_{WT}$. For case $C_{WT}$ the wind turbine is on the hilltop hence the first three profiles stop before $z = 0$.

and velocity are stronger in this case because of the flow acceleration on top of the hill. These changes in the wake shape and intensity are expected to also influence sound propagation.

The noise emitted by the turbine depends on the inflow conditions and hence varies for each case. The wind speed profile used as an input for the model is taken 10 m upwind of the turbine. The turbulent dissipation rate is set to $\epsilon = 0.01$ m$^2$.s$^{-3}$, which is a classical value for a neutral atmosphere (Muñoz-Esparza et al., 2018). The rotational speed is defined from the wind speed at hub height using the relation described in Jonkman et al. (2009). Finally, the twist of each blade segment is set to obtain an optimal angle of attack (4° for the considered airfoil) with respect to the wind speed at hub height and the rotational speed (Tian and Cotté, 2016). An equivalent overall sound power level (OASWL) can be estimated for each case. It is computed from the downwind sound pressure level in the free field according to:

$$\text{OASWL} = \text{OASPL}_{\text{ff}}(R) + 10\log_{10}(4\pi R^2) \tag{8}$$

where $R$ is the distance between the hub and the receiver taken equal to 3 km in this case. Note that for large enough $R$ (more than 1km) the OASWL becomes constant. The wind speeds are slightly different between the baseline case A ($u_{\text{hub}} = 10$ m.s$^{-1}$) and case B ($u_{\text{hub}} = 9.5$ m.s$^{-1}$). This is an effect of the hill's blocking which slightly decelerates the flow upstream. This induces a small decrease in the rotational speed $\Omega$ and in consequence a slight decrease of 0.6 dBA in OASWL. Case C on the other hand shows a significant increase in wind speed (25% compared to $A_{WT}$) and rotational speed (8% compared to $A_{WT}$) due to the elevated hub and the increased wind speed at the top of the hill. Consequently, the OASWL increases by 2 dBA compared to case A. These results are summarized for each case in Table 1.

## 4 Numerical set-up for the propagation simulations

For each case presented in Sec. 3, numerical simulations of the noise propagation are performed using the LEE method described in Sec. 2.2. The moving frame has a length of 300 m×300 m. The grid step is set to $\Delta x = \Delta z = 0.05$ m and the CFL to



**Table 1.** Wind turbine source parameters calculated for each case.

|  | $u_{\mathrm{hub}}$ (m.s$^{-1}$) | $\Omega$ (rpm) | OASWL (dBA) |
|---|---|---|---|
| case A | 10.0 | 11.2 | 97.7 |
| case B | 9.5 | 10.9 | 97.1 |
| case C | 12.4 | 12.1 | 99.6 |

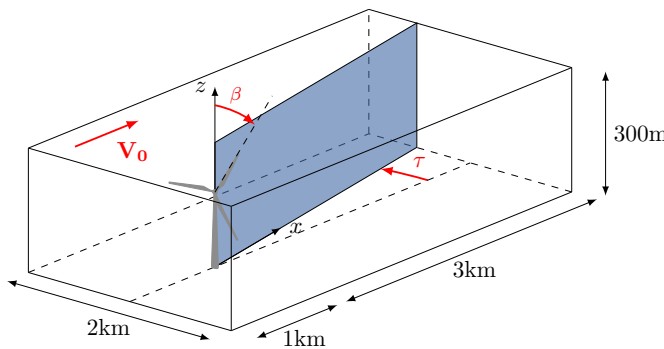

**Figure 7.** Sketch of the computational domain. $\beta$ is the blade angle and $\tau$ is the propagation angle.

0.5. Those numerical parameters produce accurate results up to 1 kHz. The results are then computed in the frequency domain and presented averaged in third-octave bands.

First, the simulations are performed in a 3D domain using a $N \times 2D$ approach. The sound propagation is computed in vertical planes as described in Sec. 2, neglecting transverse propagation effects. The 2D simulations are performed for a set of angles of propagation $\tau$ (Fig. 7), to compute a $\Delta L$ map around the wind turbine. The flow is almost symmetrical with respect to the $x$ axis, hence only angles between 0° and 180° are considered. This set of simulations is computed only for the source at the hub height (100 m) and for a wind speed $u_{\mathrm{ref}} = 8$ m.s$^{-1}$. The 3D rectangular domain has a size of 4 km×2 km×300 m as illustrated in Fig. 7. To capture all significant propagation effects, sound propagation is simulated up to 3 km downwind and up to 1 km in the crosswind and upwind directions. Hence, the angular step must be smaller for the computations downwind of the wind turbine. For $\tau$ between 0° and 15°, a 2° angular step is used, and an angular step of 5° is used for the rest of the domain. This is done to save computational resources as a 1 km propagation simulation takes around 240 CPU hours, and a full simulation (with all propagation angles) takes around $2 \times 10^4$ CPU hours. The $\Delta L$ fields obtained from these simulations are presented in Sec. 5.1.1.

A second set of simulations is performed only in the downwind ($\tau = 0°$) and upwind ($\tau = 180°$) directions for the six cases with $u_{\mathrm{ref}} = 10$ m.s$^{-1}$. Here, several source heights are considered to compute the OASPL and AM using the MM approach (see section 2). The wind turbine considered is the same as in Colas et al. (2023). Seven source heights are considered for all cases except for case $\mathrm{C_{WT}}$ in the downwind direction where 30 source heights are used to reach convergence of the AM



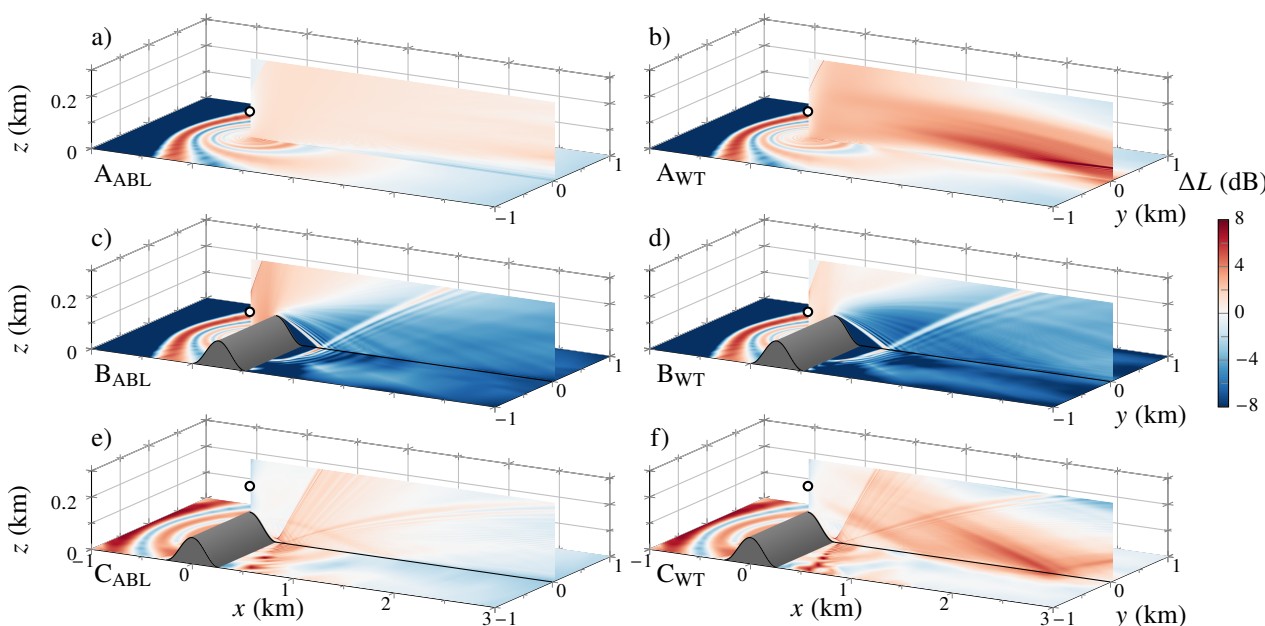

**Figure 8.** $\Delta L$ field at $f_c = 800$ Hz for the six cases studied (b, d, and f) with and (a, c, and e) without the wake in the mean flow. The source is at $x = y = 0$ and $z = 100$ m (circle) and the wind speed at 100 m is equal to 8 m.s$^{-1}$.

results. The comparison between the two wind speeds is shown in Sec. 5.1.2 and the OASPL and AM obtained from the six cases are compared in Sec. 5.2.

## 5  Results and Discussion

### 5.1  Propagation effects

#### 5.1.1  General case description

Fig. 8 shows $\Delta L$ fields for the six considered cases at $f_c = 800$ Hz on the planes $y = 0$ and $z = 2$ m. The reference cases without wind turbines are on the left and the results with wind turbines are on the right. For all cases, a point source is placed 100 m above the ground, which corresponds to the hub height. Without the wind turbine and with flat ground (Fig. 8a), the flow varies only along the vertical direction. Upwind it corresponds to a negative effective sound speed gradient that is responsible for the shadow zone observed in cases $A_{ABL}$ and $A_{WT}$. Sound waves are refracted upwards as they propagate leading to a zone with very low SPL at the ground for $x < -500$ m. The positive effective sound speed gradient downwind refracts sound waves downward and can lead to an increase in SPL at the ground (Barlas et al., 2018; Heimann and Englberger, 2018). For case $A_{ABL}$ (Fig. 8a downstream of the turbine), the gradient does not produce strong focusing because the velocity gradient is small for a neutral ABL.





The velocity gradient induced by the wake (case $A_{WT}$) has a significant effect on sound propagation (Fig. 8b). As shown in previous studies by Barlas et al. (2017b) and Heimann and Englberger (2018), the wake from a wind turbine has a significant impact on the SPL observed at ground level. This effect is clearly visible when comparing Figs 8a and 8b. The wake, created by the wind turbine's downstream velocity deficit, acts as a waveguide that causes sound waves to focus at a point approximately 3 kilometers away from the noise source. Note that this focusing effect occurs for small propagation angles only ($\tau \approx 0$) and that for higher angles of propagation, the $\Delta L$ field is similar to the one without the wake. This emphasizes the importance of considering the wake when studying sound propagation in such scenarios.

The hill impacts sound propagation in two ways: directly, by causing reflections and diffraction of the sound waves due to the terrain, and indirectly, by altering the average airflow patterns. Hence, the effect of the wake presented for cases $A_{ABL}$ and $A_{WT}$ can either be strengthened or counterbalanced by the influence of the terrain. In the case of the wind turbine positioned upstream of the hill (Figs. 8c, 8d), most of the propagation effects can be attributed to the hill itself, and the wind turbine wake has a minimal impact. The primary effect is the shielding effect of the hill that creates a shadow zone downwind. A secondary effect is the $\Delta L$ amplification that can be observed at $x = 700$ m at ground level. It is caused by the strong velocity gradient created by the hill's wake. However, even with this refraction by the mean flow, the shielding effect of the hill on sound propagation is strong. Hence, $\Delta L$ is negative after the hill, except for this focusing zone. Cases $B_{ABL}$ and $B_{WT}$ present the same upwind shadow zone as in cases $A_{ABL}$ and $A_{WT}$, which shows that the hill has almost no influence on the upwind propagation. The velocity gradient induced by the wake itself is small in comparison to the hill's wake (see Fig. 5b). Hence, the waveguide effect that was observed for case $A_{WT}$ is not visible. This indicates that the effect of the wake created by a turbine placed upstream of a hill is overshadowed by the effect of the hill itself. Note that a stronger effect of the turbine's wake is visible when considering increased source heights.

When the wind turbine is located on the hilltop (Figs. 8e, 8f), the geometry of the hill first creates a cusp caustic just above the bottom of the hill (at $x = 260$ m). The caustic then separates into two branches, one going directly up and one reflecting towards the ground at 330 m. This is only an effect of the terrain and can be observed for both cases $C_{ABL}$ and $C_{WT}$. On the top view, the caustic branch hitting the ground moves further away from the hill as the propagation angle increases. This is equivalent to propagation over a hill with a smaller slope angle, for which the caustic is created higher and the downward refracted branch hits the ground further away from the hill's base. The second focusing effect visible for case $C_{ABL}$ (Fig. 8e) is similar to what is observed in case B. The diminished wind speed behind the hill creates a focal area at the ground positioned at $x = 1300$ m. However, this effect is less noticeable than in case B due to the more pronounced direct influence of the source. For case $C_{WT}$ (Fig. 8f) the wind turbine wake induces different focusing patterns somewhat similar to what is visible in case $A_{WT}$. Note that the focusing patterns induced by the flow and hill depend on the source height and wind velocity. Hence they are further studied in the following sections.

### 5.1.2 Effect of wind speed

In this section, the effect of the velocity on propagation is briefly examined. The previous results were obtained for a wind profile characterized by a wind speed at 100 m equal to $u_{ref} = 8$ m.s$^{-1}$. In general, a wind turbine operates at wind speeds



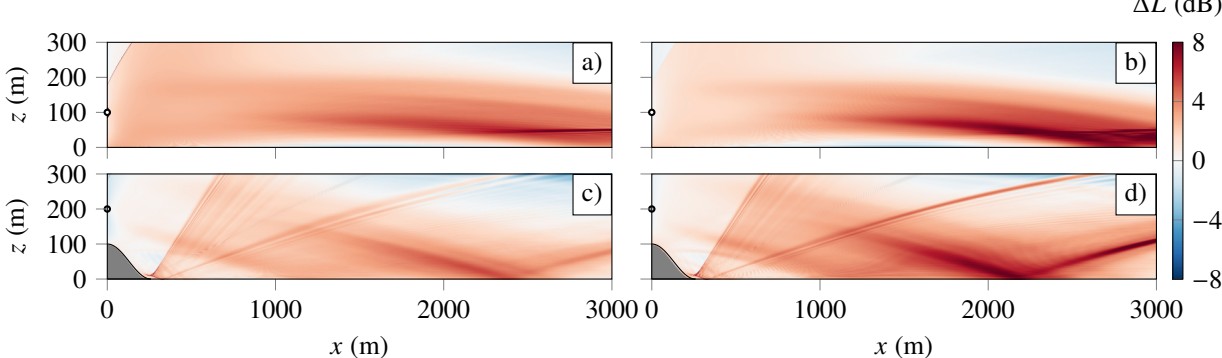

**Figure 9.** $\Delta L$ field at $f_c = 800$ Hz for case (a, b) $A_{\mathrm{WT}}$ and (c, d) $C_{\mathrm{WT}}$, with (a, c) $u_{\mathrm{ref}} = 8$ m.s$^{-1}$ and with (b, d) $u_{\mathrm{ref}} = 10$ m.s$^{-1}$. The source is at 100 m from the ground (circles).

between 6 m.s$^{-1}$ and 25 m.s$^{-1}$ (Jonkman et al., 2009). Hence, additional simulations are run for case $A_{\mathrm{WT}}$ and $C_{\mathrm{WT}}$ with $u_{\mathrm{ref}}$ equal to 10 m.s$^{-1}$ to assess the effect of the wind speed on the refractions. These new cases are made from the available LES data by scaling the flow fields. The $\Delta L$ contour plot at 800 Hz for cases $A_{\mathrm{WT}}$ and $C_{\mathrm{WT}}$ are compared for both wind speeds in Fig. 9. The general dynamics are similar for both wind speeds but the focusing effect is enhanced for $u_{\mathrm{ref}} = 10$ m.s$^{-1}$. With

flat terrain, the steeper vertical velocity gradient guides sound waves more efficiently toward the ground resulting in a greater $\Delta L$, especially between 2500 and 3000 meters.

  The $\Delta L$ for receivers at 2 m above the ground are presented for the two wind speeds in Fig. 10. The wave focusing created by the wake induces a maximum after $x = 2000$ m in the $\Delta L$ values for all cases. The effect of the caustic at the bottom of the hill is also present for case C with a strong peak at $x = 300$ m. The change in wind speed has a similar effect on $\Delta L$ for

case $A_{\mathrm{WT}}$ and case $C_{\mathrm{WT}}$. By increasing the wind speed the focusing phenomenon intensifies: the peak is stronger and more localized. There is a 1.5 dB increase between $u_{\mathrm{ref}} = 8$ m.s$^{-1}$ and $u_{\mathrm{ref}} = 10$ m.s$^{-1}$ for case $A_{\mathrm{WT}}$, and a 0.5 dB increase for case $C_{\mathrm{WT}}$. As the wind speed increases, the $\Delta L$ upstream of the focusing zone decreases for case $A_{\mathrm{WT}}$ (between $x = 1000$ m and $x = 1800$ m). This is explained by the fact that less energy is redirected outside of the focusing area as the amplification becomes stronger and more localized. The focusing zone on the ground also appears closer to the source. For case C, the

maximum shifts 250 m closer to the source at 10 m.s$^{-1}$ compared to the case at 8 m.s$^{-1}$. A small shift in the focusing induced by the hill's wake is also noticeable at $x = 700$ m. On the opposite, the focusing induced by the hill geometry at $x = 520$ m does not appear to be shifted by the change in wind speed.

  To summarize the above, with increasing wind speed, the focusing effect intensifies and moves closer to the source where sound pressure attenuation due to geometrical spreading and atmospheric absorption is lower. The wind turbine noise produc-

tion also increases at higher wind speeds. Both effects lead to an increase in SPL. Hence, in the following only the case where $u_{\mathrm{ref}} = 10$ m.s$^{-1}$ is studied.



WIND
ENERGY
SCIENCE
DISCUSSIONS

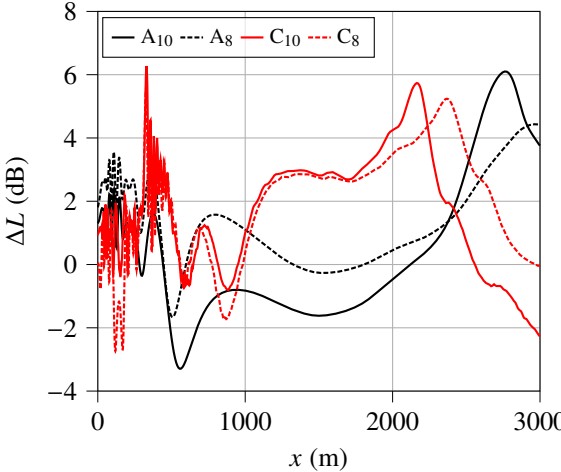

**Figure 10.** $\Delta L$ at 2 m height at $f_c = 800$ Hz for cases $A_{WT}$ and $C_{WT}$, for $u_{ref} = 8$ m.s$^{-1}$ (dashed lines) and $u_{ref} = 10$ m.s$^{-1}$ (solid lines)

### 5.1.3 Effect of source height

The influence of the source height on sound propagation is crucial for the prediction of wind turbine noise. In the model considered in this work, the rotational motion of the blades is translated into a vertical motion of the sources. Hence, sound
propagation depends on the source position, which impacts locations where sound amplification at the ground is observed. In addition, the position of the focusing zone changes with source height. This section aims to gain some insights into the different propagation effects and their dependence on source heights.

The $\Delta L$ computed for case $A_{WT}$ for three source heights is shown in Fig. 11a,b and c; Fig. 11d shows the $\Delta L$ for a line of receivers at $z = 2$ m for the seven source heights considered. Three different focusing effects can be observed. For a
source height $h_s = 58$ m (Fig. 11a), the sound waves are first refracted upwards by the bottom of the wake, then, after the wake recovery, the sound waves are redirected towards the ground by the positive vertical velocity gradient in the ABL. For $h_s = 100$ m (Fig. 11 b), two focusing zones can be observed: one at the bottom of the wake and one at the top. They reach the ground after 2.5 km leading to an increase over a larger surface at the ground. This is similar to what was found in Barlas et al. (2017b). Finally, for $h_s = 142$ m (Fig. 11 c), only one focusing zone created by the top wake gradient is observed. Here, the
top-wake positive gradient effectively redirects sound waves toward the ground. Fig. 11d shows the differences in SPL at the ground induced by these focusing patterns. For the highest source, a clear peak is present at $x = 1700$ m. As the source height decreases the peak gets wider, is less pronounced and moves downstream. Finally, when the source is too close to the ground the focusing occurs at a higher altitude and does not reach the ground before $x = 3000$ m.

The effect of the flow and in particular the wake on sound propagation appears to strongly depend on the source heights.
To validate and provide additional explanation for the findings in Fig. 11, a ray tracing simulation is conducted (Candel, 1977; Scott et al., 2017). The ray paths are shown superimposed on the velocity fields in Fig.12 for elevation angles $\varphi$ between -20°





and 20°. The flow gradient bends the rays leading to focal areas called caustics. These caustic curves (shown in black dashed lines) delimit a region in which the number of rays increases. The caustic curve itself corresponds to high sound pressure locations, as the number of rays increases towards infinity (in the geometrical approximation). First, note that both prediction
methods give identical results for the position of the focusing zones. The caustics positions in Fig. 12 match with regions of high $\Delta L$ levels in Fig. 11.

In addition, the ray-tracing approach provides information on the path taken by the sound waves leading to this increase in $\Delta L$. For the highest source (Fig. 12 c) the rays launched at small positive angles are strongly redirected toward the ground by the positive shear at the top of the wake. As the source moves down the rays launched upwards travel more distance before being
redirected downwards by the wake, hence the focusing zone reaches the ground farther from the source. For sources sufficiently low (Fig. 12a and b) the rays launched downwards are first refracted upwards by the bottom-wake negative gradient before being refracted downwards by the positive ABL shear. For $h_s = 100$ m (Fig. 12b) both phenomena occur. The rays launched upwards are downward-refracted by the top-wake positive gradient and the rays launched at negative angles are redirected towards the ground but less efficiently due to the bottom-wake negative velocity gradient. Note that this method allows us to

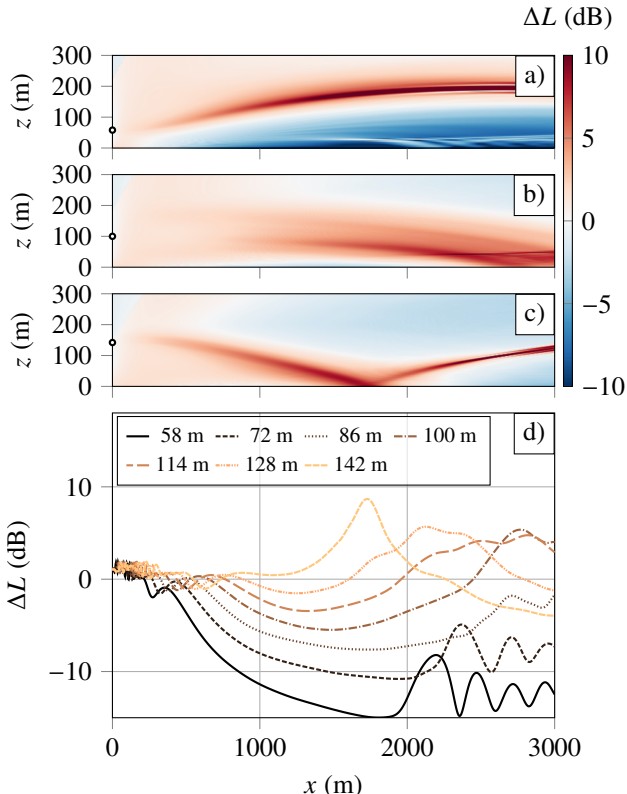

**Figure 11.** $\Delta L$ field at $f_c = 1000$ Hz for case $A_{WT}$ for three different source heights: a) $h_s = 58$ m, b) $h_s = 100$ m, c) $h_s = 142$ m; d) $\Delta L$ at 2 m above the ground for the seven source heights.





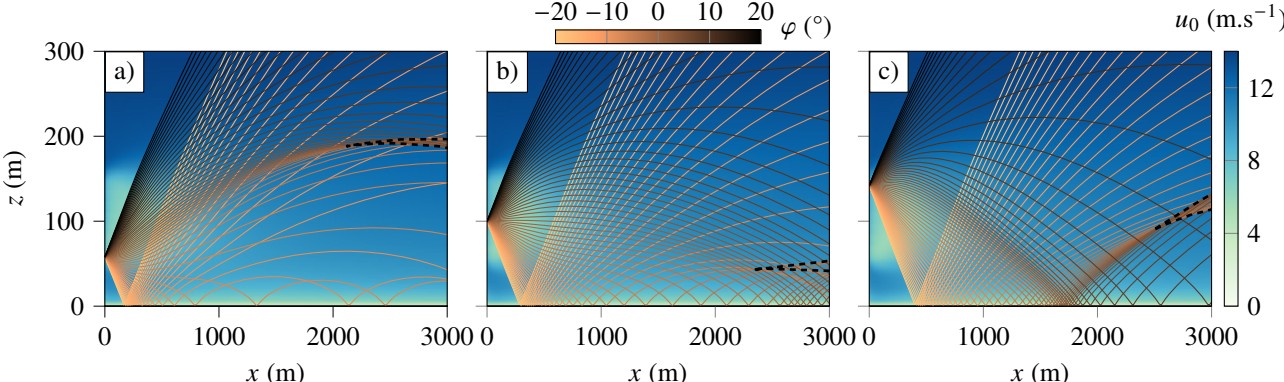

**Figure 12.** Atmospheric refraction in case $A_{WT}$ displayed using a ray-tracing method for three different source heights: a) $h_s = 58$ m, b) $h_s = 100$ m, c) $h_s = 142$ m. The rays are superimposed over the wind speed fields $u_0$, $\varphi$ is the initial elevation angle of each ray, and caustics are shown in black dashed lines.

precisely show the refraction effect of the wind shea, but has limitations when it comes to computing precise SPL as it is a high-frequency approach.

The same type of analysis can be performed for case $C_{WT}$. The $\Delta L$ fields are again plotted for three source heights in Fig. 13a,b,c. In Fig. 13d the $\Delta L$ values are plotted for 7 source heights ranging from $h_s = 152$ m to $h_s = 248$ m. The source heights 215 m, 230 m, and 248 m correspond to the situation shown in Fig. 13c for which only one focusing zone is present

similarly to case $A_{WT}$. With decreasing source height, the focusing zone shifts away from the source. When the sources are located at 185 m and 200 m two distinct peaks appear; which correspond to the two focusing zones visible in Fig. 13b. These two peaks start to merge as the source approaches the ground (see Fig. 13a). Consequently, for source heights between 152 m and 176 m, the $\Delta L$ at the ground shows a more complex pattern because the two focusing zones are superimposed. The cusp caustic at the bottom of the hill is observed for the three source heights in Fig. 13a,b and, c. The corresponding increase in $\Delta L$

is also visible in Fig. 13d between $x = 250$ m and $x = 500$ m. These peaks move closer to the base of the hill for sources close to the ground.

To better understand the source height influence on the focusing pattern, the ray-tracing results are again shown in Fig. 14 for the three source heights. For $h_s = 170$ m two caustic branches match the $\Delta L$ increase observed in Fig. 13 a. The wake induces a convergent beam of rays creating a cusp caustic. This caustic may correspond to the one observed in case $A_{WT}$ in

Fig. 12, but here, due to the geometry of the wake, the branches hit the ground before 3 km. As the source height increases (Fig. 14 b) two focusing zones appear. In this case, the rays launched toward the ground are not refracted upward anymore but are only deviated by the bottom of the wake. There is no caustic before the bottom-wake focusing hits the ground. The second focusing zone is induced by the top-wake gradient. Finally, for $h_s = 248$ m (Fig. 13 c) the behavior is similar to what is shown for case $A_{WT}$ where the rays are strongly refracted by the wake gradient at the top and only this effect dominates.



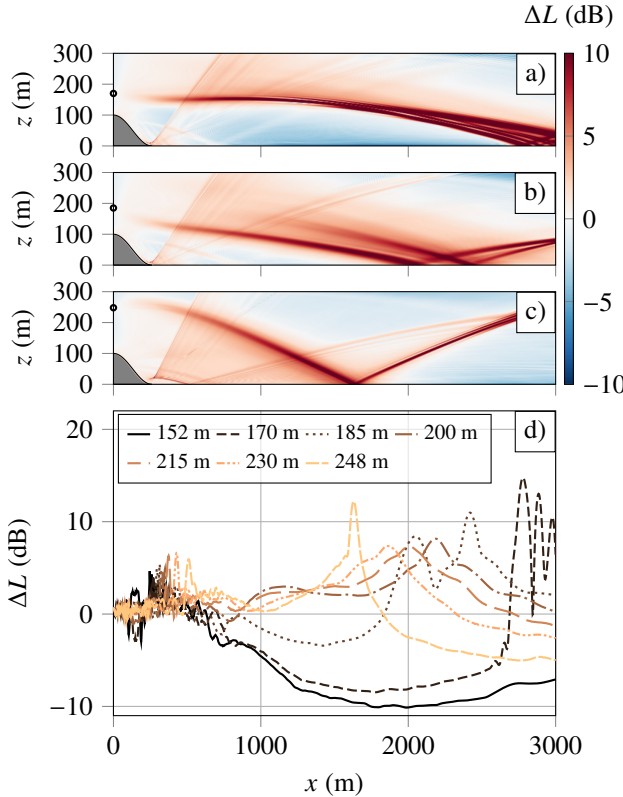

**Figure 13.** $\Delta L$ field at $f_c = 1000$ Hz for case $C_{WT}$ for three different source heights: a) $h_s = 170$ m, b) $h_s = 185$ m, c) $h_s = 248$ m; d) $\Delta L$ at 2 m above the ground for seven source heights.

Through the analysis of cases $A_{WT}$ and $C_{WT}$, we saw that the wind turbine wake has a strong influence on the propagation. More precisely, the focusing pattern is modified by the source heights because sound waves are refracted differently by the wind turbine wake. The effect of the source height on case $B_{WT}$ is not presented here as the focusing induced by the wind turbine wake in this case hits the ground far away from the source with low intensity. Nevertheless, the amplification position moves with source height, as for case $A_{WT}$ and $C_{WT}$, inducing some AM in the far field, as presented in the following section.

## 5.2   Wind turbine noise

In this section, the OASPL due to the wind turbine is analyzed. We start by discussing the OASPL results from a single blade. Next, we compare the OASPL, averaged over one full rotation, and AM for a wind turbine with 3 blades for all six cases.

### 5.2.1   One blade OASPL

First, snapshots of the OASPL obtained for one blade for case $C_{WT}$ are presented for six different blade angles in Fig. 15. For
all the following plots the zone where the Amiet's model is not valid ($|x| < 10 \times c_0/f_{\min} = 70$ m) is shown with a gray area.



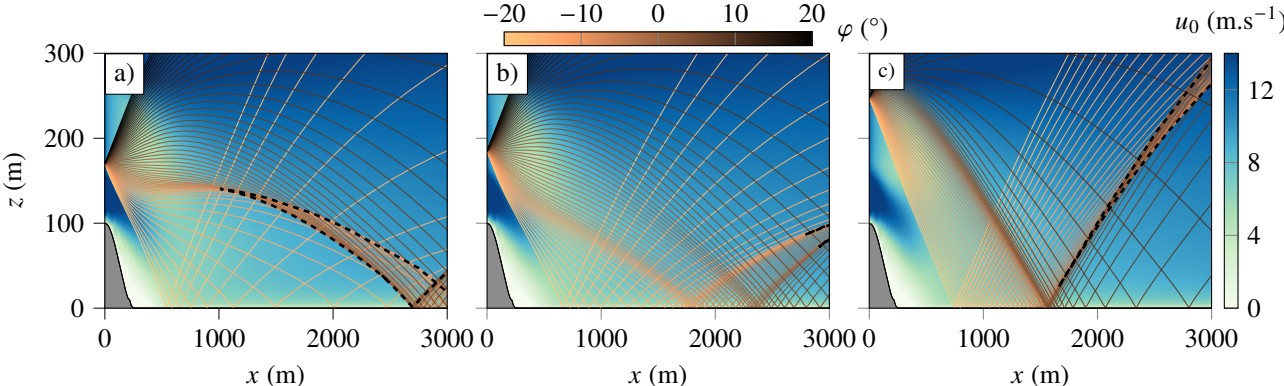

**Figure 14.** Atmospheric refraction in case $C_{WT}$ displayed using a ray-tracing method for three different source heights: a) $h_s = 170$ m, b) $h_s = 185$ m, c) $h_s = 248$ m. The rays are superimposed over the wind speed fields $u_0$, $\varphi$ is the initial elevation angle of each ray, and caustics are shown in black dashed lines.

The effect of the source model and the propagation are visible. The OASPL is higher close to the source and decreases as the receiver moves further away, due to geometrical spreading and atmospheric absorption. In addition, the flow and topography effects described in the previous section are still visible. The dependence of OASPL on blade orientation comes from the activation of different $\Delta L$ fields (Sec.5.1.3) as the blade moves up and down. The blade starts upwards on the snapshot at the

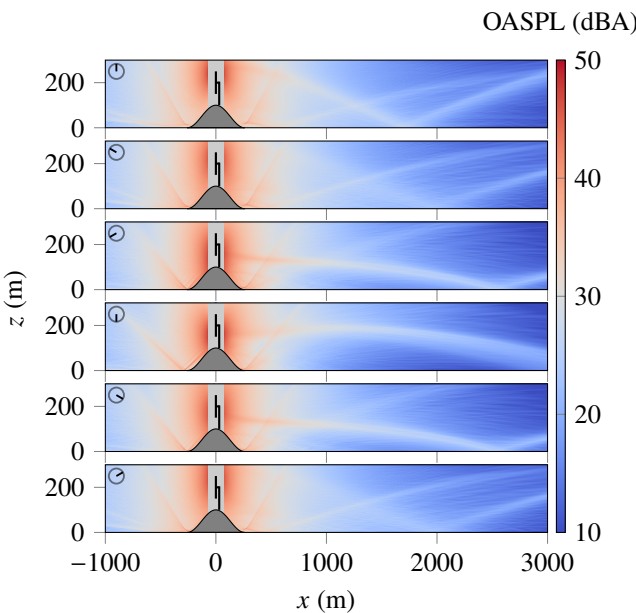

**Figure 15.** OASPL for case $C_{WT}$ for one blade at different angular positions (indicated on the top left corner).



top, with the focusing hitting the ground at $x = 1800$ m. Then the blade moves downwards to $\beta = 180°$ and the focusing zone reaches the ground at around $x = 3000$ m. The blade finally moves upwards again and the focusing zone moves back towards the source. The source height also influences the OASWL, as the wind speed is higher at $z = 148$ m than at $z = 52$ m. This evolution of OASWL is responsible for AM close to the source while the change in propagation path induces AM in the far field as the peak positions at the ground are modified. This effect was shown in Heimann et al. (2018) with the particles emitted

from the lower sources being refracted upward and those emitted from the highest sources being refracted downwards. These phenomena are also visible for cases $A_{WT}$ and $B_{WT}$ and will be shown in the next section.

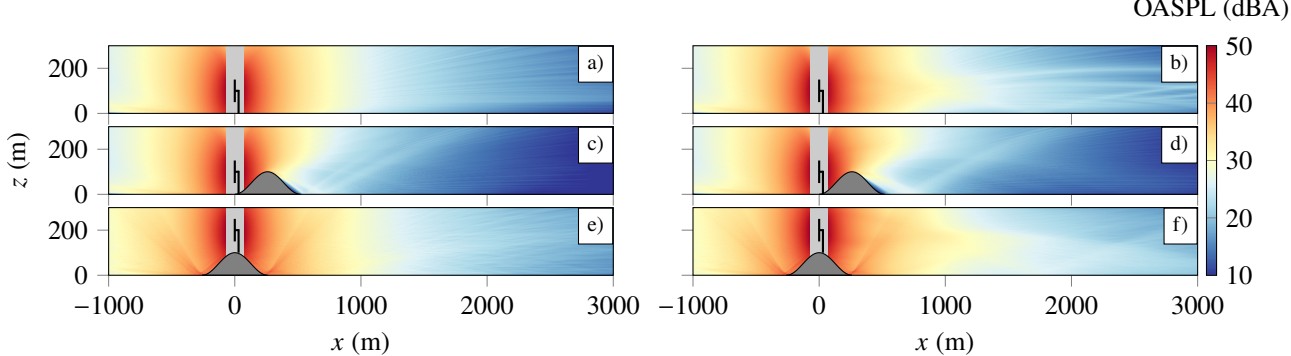

**Figure 16.** Averaged OASPL over one rotation for the six cases with and without the wake. a) $A_{ABL}$, b) $A_{WT}$, c) $B_{ABL}$, d) $B_{WT}$, e) $C_{ABL}$, f) $C_{WT}$.

### 5.2.2 Averaged OASPL and AM

The OASPL averaged over one rotation are shown in Fig. 16 for the six cases and a velocity at $z = 100$ m of 10 m.s$^{-1}$. By summing the contribution of the three blades and averaging over one rotation the focusing effects previously described tend to

average out for all cases. The OASPL does not increase downwind when the wind turbine wake is not in the flow (Figs. 16a, c, and e). The shadow zone is distinguishable upwind close to the ground for cases $A_{ABL}$ and $B_{ABL}$. Fig. 16c shows the focusing zone induced by the hill's wake in case $B_{ABL}$. Finally, in Fig. 16e, caustics at the bottom of the hill in case $C_{ABL}$ are present on both sides. As previously shown, focusing patterns appear downwind for cases accounting for the wake (Figs. 16b,d,f). Even for case $B_{WT}$, where the shielding effect of the hill is strong, sound waves appear to be redirected toward the ground at a great

distance. Hence, an increase in OASPL downwind at the ground is expected for these cases.

Corresponding AM fields are shown in Fig. 17. The AM is very low for all cases without the wake. It can be concluded that downwind and upwind AM is not significantly affected by the OASWL variation due to blade rotation. Other authors have reached this conclusion (Cotté, 2019). They would increase crosswind as the source moves closer and farther from the receiver (Cotté, 2019; Barlas et al., 2017a; Mascarenhas et al., 2023). The AM increases close to the ground for $x < -500$ m for cases

A and B due to the upwind shadow zone. For cases $B_{WT}$ and $B_{ABL}$, the zone with large AM after the hill corresponds to the



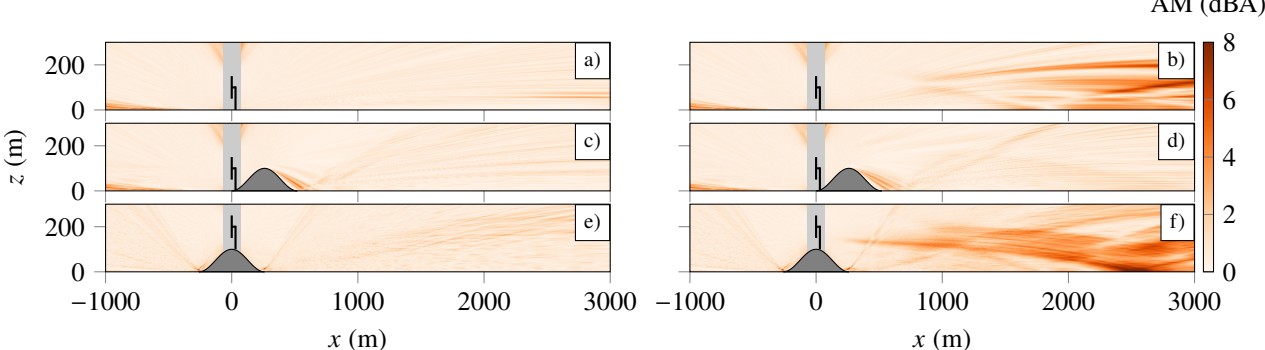

**Figure 17.** AM for the six cases with and without the wake. a) $A_{ABL}$, b) $A_{WT}$, c) $B_{ABL}$, d) $B_{WT}$, e) $C_{ABL}$, f) $C_{WT}$.

focusing induced by the hill's wake. An important result is that no AM is present downwind for the cases without the wake (Figs. 17a,c,e). On the other hand, the variation in focusing patterns in cases $A_{WT}$ and $C_{WT}$ leads to strong AM downwind (Figs. 17b,f).

The comparison of the average OASPL is presented in Fig. 18a for receivers at 2 m height. Case $A_{ABL}$ shows a downwind
decrease mostly due to the atmospheric absorption and the geometrical decay. Upwind, the shadow zone becomes discernible at $x = -700$ m with a steeper decrease. The downwind OASPL is higher in case $C_{ABL}$ with a consistent difference of 5 dBA with case $A_{ABL}$. Note that 2 dBA are due to the increase in OASWL, the remaining 3 dBA come from propagation effects. Upwind, the shadow zone moves away from the source as the wind turbine is higher. For case $B_{ABL}$ the shielding of the hill downwind induces a strong dip between 250 m and 600 m. However, the OASPL at the focus zone ($x = 700$ m) is of
comparable amplitude with the SPL of the other cases at this position. Hence, although the hill shields the noise, levels remain significant at ground level just after the shadow zone. Downwind of the focusing zone, the levels are 4 dBA lower than for case $A_{ABL}$. In this case, 0.6 dBA are lost at noise emission. In the upwind direction, ($x < 0$), case A and case B are almost equivalent as there is no effect of the topography or the mean flow on sound propagation.

The main effect of the wake on the OASPL is an increase downwind at ground level. The wake seems to have a similar effect
on the average OASPL for cases $A_{WT}$ and $C_{WT}$. The average OASPL increases by 4 dBA for $x > 1700$ m, which corresponds to the closest focusing zone shown in Fig. 11 and Fig. 13. For case $B_{WT}$, a similar increase can be observed, but farther from the source ($x > 2200$ m). It is worth noting that these results differ from what is obtained when using a point source approximation where the focusing zone is more localized (Colas et al., 2022). In this previous study, it was found that case $A_{WT}$ could show higher OASPL in the far field than case $C_{WT}$.

Finally, the corresponding AM values are presented in Fig. 18b. Upwind there is almost no influence of the wake. For cases A and B a clear increase in AM at $x = -700$ m is visible. It corresponds to the start of the shadow zone. At this distance, the receiver moves in and out of the shadow zone as the blades move up and down hence yielding AM (Cotté, 2019). For case C, the shadow zone starts further away due to the increased source height. The peaks in AM at $x = 260$ m and $x = -260$ m are



created by the caustics at the bottom of the hill. Downwind, it is clear that for all cases the AM levels are much higher when
the wake is present. This is similar to what was found in (Heimann et al., 2018; Barlas et al., 2017b). The case $A_{WT}$ presents
an AM increase up to 4 dBA with maxima at around $x = 1700$ m and $x = 2200$ with a strong dip at $x = 1900$ m. This shape is
a consequence of the superposition of the three blades. As one blade rotates, the focusing zone moves farther and closer to the
source. But because of the contribution of the three blades, there is always an amplification at $x = 1900$ m corresponding to
a source height around hub height. This creates a zone where the OASPL does not vary and hence where the AM is low. The
same behavior is visible for case $C_{WT}$ with even stronger AM (up to 9 dBA). Two dips are distinguishable in this case, as it
was shown that two different focusing zones are created when the source height is around hub height (see Sec. 5.1.3). For case
$B_{WT}$, AM is also present in the far field but does not reach values higher than 2 dBA. For case B another zone of AM is visible
at $x = 600$ m for both case $B_{WT}$ and $B_{ABL}$. This increase by 4 dBA comes from the refraction induced by the hill's wake.

We recover some well-known effects of wind turbine sound propagation, such as downwind amplification induced by the
wind turbine wake and upwind AM created by the negative effective sound speed gradient Mascarenhas et al. (2023); Barlas
et al. (2017b); Heimann et al. (2011). With the turbine on a hilltop both the noise levels and the AM are greatly increased
downwind compared to the flat terrain case. This could result in increased annoyance and is a combined effect of the hill and
wind turbine wake. With the wind turbine upstream of the hill, a strong shielding effect reduces the noise level downwind.
Nonetheless, levels comparable to the baseline flat case were found just after the hill because of refraction induced by the hill's
wake.

## 6 Discussion

In this section, we aim to compare our results with those from previous numerical studies. For the flat terrain case, our conclu-
sion is in good agreement with the results from Barlas et al. (2017b). The effect of wind speed and source height is very similar
in both studies. The increase in wind speed brings the amplification zone closer to the source and the peaks are sharpened
(see Fig. 6 in (Barlas et al., 2017b)). Likewise, the source height impacts the refraction downwind, and the focusing zone is
observed closer to the turbine for a higher source (see Fig. 9 in (Barlas et al., 2017b)). The effect of the wake on the OASPL
downwind of the turbine is also very similar. They found that the wake increases by 5 dB the average OASPL for $x > 1000$ m
and $z = 2$ m (see Fig. 13 in (Barlas et al., 2017b)). This is slightly closer to the source than our results (the increase starts at
around $x = 1700$ m), but this can be explained by the increased wind speed at hub height and the smaller wind turbine in their
study.

Barlas et al. (2018) also studied AM evolution with distance for varying atmospheric stability. For the neutral case, they
found an increase in AM downwind with a maximum just before $x = 1500$ m. Again, this is slightly closer to the source
than what we found, but this can also be attributed to the wind turbine being smaller in their study. They also found several
maximums associated with different refraction patterns induced by different blade positions. The dip in the AM pattern is
less pronounced in their study because they used an unsteady approach. AM is not equal to zero in this area, contrary to our
findings, as the flow and hence the propagation path vary even when the blade position is identical.

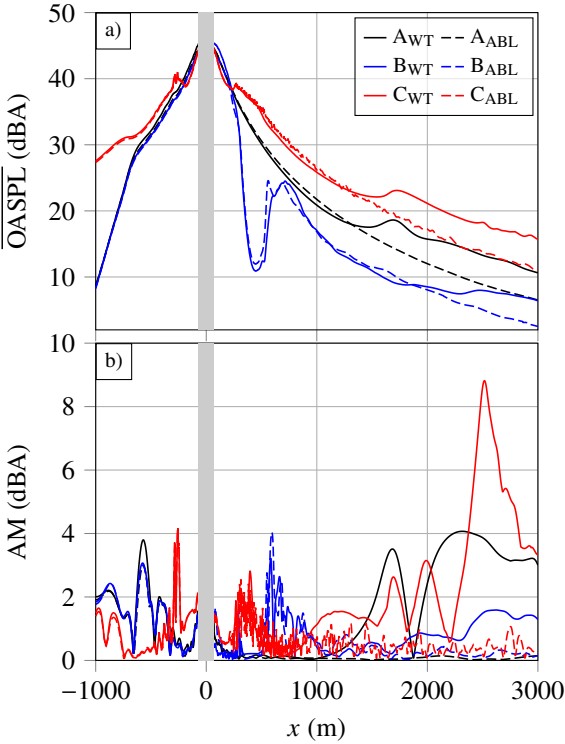

**Figure 18.** Comparison of a) the OASPL and b) the AM at 2 m above the ground between the six cases.

Heimann et al. (2011) showed the importance of 3D sound propagation in the context of wind turbine noise. Two important effects are not accounted for in our study. The spanwise wind speed gradient creates sound wave refraction in the horizontal direction, which is not represented in our model. The wind veer also impacts propagation inducing an asymmetric SPL distri-
bution. Despite these differences, the two studies can be compared for a flat terrain scenario. They found an increase in average OASPL starting for $x > 1000$ m with various patterns depending on atmospheric stability. This is closer to the source compared to our results, even though the wind turbine dimensions and wind speed at hub height are the same. The use of different wind speed profiles and wake models could explain this discrepancy.

Our results for a wind turbine on a hilltop also differ from those reported in Heimann et al. (2018). In their study, the hill
and wake effects tend to compensate for each other, while in our work the highest OASPL is obtained for the case with the hill and the wind turbine wake accounted for. There are several explanations for these two opposite conclusions. First, their study does not account for the increased sound emission while, in our case, the OASWL is higher for case $C_{WT}$. But even if changes in sound emission are neglected, we saw that propagation effects contribute to an increase in SPL compared to the flat terrain case. However, it is difficult to assess which propagation effect is responsible for this difference. The propagation
distance considered in Heimann et al. (2018) is much smaller and most of our conclusions are based on results farther than 1 km downwind. Additionally, the hill height in our study is larger. However, due to atmospheric refraction, this does not necessarily





imply that the zone of amplification would be located further from the source. Finally, the wake shape is different between the two studies and, despite having a similar effect, it is not clear if the focusing zone should be different.

## 7  Conclusions

This study demonstrates that terrain topography can significantly affect wind turbine sound propagation. We used linearized Euler equations, solved in a moving computational frame, and an extended source model to simulate sound propagation from a wind turbine in the presence of topography. The combined effect of a 2D hill, *i.e.* a ridge, and the velocity gradient created by the terrain and the wind turbine was studied for three configurations. First, we recover some well-known effects of wind turbine sound propagation such as downwind focusing induced by the wind turbine wake and upwind amplitude modulation

created by the vertical velocity gradient in the ABL. In the case of a wind turbine on a hilltop, both the sound levels and the AM are greatly enhanced downwind compared to the flat case. This could result in increased annoyance and is mainly an effect of mean flow field modification. In all cases, the far-field AM downwind is created by the wind turbine's wake. When the turbine is placed in front of the hill, we observed that, despite a strong shielding effect from the terrain, SPL just after the hill is comparable to that of the flat case. This is due to focusing induced by the hill's wake which also increases AM at this

location. Hence, it would be possible to encounter cases where annoyance issues are raised despite a supposed shielding effect of topography. In this case, the effect of the wind turbine wake is limited as sound propagation is mainly determined by the geometry of the terrain and the flow around the hill.

Furthermore, we saw that topography also affects sound emission. For a turbine on a hilltop, the sound emitted is higher than in the flat case due to increasing wind speed. For a turbine upstream of a hill, the sound emitted is slightly lower due

to the decrease in wind speed before the hill. Finally, we find that the sound-focusing effect becomes more pronounced and closer to the turbine with increasing wind speed and that most of the propagation effects are directly behind the wind turbine (close to $\tau = 0$). However, this might be because our study used a N$\times$2D approach, which doesn't consider the full 3D sound propagation through the wake. Also, turbulence in the atmosphere can scatter the sound and reduce the focusing effect we observed, but further research is needed to determine to what extent this happens.

*Data availability.* Data from numerical simulations are available from the authors on reasonable request.

*Author contributions.* J. Colas did the main research work, performed and analyzed the numerical simulations and wrote most of the paper. The propagation model was implemented by J. Colas, the source model was developed by B. Cotté and a version was implemented by J. Colas. The LES code was developed by R. Stevens and his team at Twente University. Through discussions and feedback, all authors contributed to the interpretation and discussion of the results. The paper was revised and improved by all authors.





*Competing interests.* The authors declare that no competing interests are present.

*Acknowledgements.* The authors thank Luoqin Liu for providing access to the LES data of Liu and Stevens (2020). This work was performed within the framework of the LABEX CeLyA (ANR-10-LABX-0060) of Université de Lyon, within the program "Investissements d'Avenir" (ANR-16-IDEX-0005) operated by the French National Research Agency (ANR). The authors were granted access to the HPC resources of PMCS2I (Pôle de Modélisation et de Calcul en Sciences de l'Ingénieur et de l'Information) of Ecole Centrale de Lyon, PSMN (Pôle
Scientifique de Modélisation Numérique) of ENS de Lyon and P2CHPD (Pôle de Calcul Hautes Performances Dédiés) of Université Lyon I, members of FLMSN (Fédération Lyonnaise de Modélisation et Sciences Numériques), partner of EQUIPEX EQUIP@MESO. This work was supported by the Franco-Dutch Hubert Curien partnership (Van Gogh Programme No. 49310UM). For the purpose of Open Access, a CC-BY public copyright license has been applied by the authors to the present document and will be applied to all subsequent versions up to the Author Accepted Manuscript arising from this submission.



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
