# Peer review of "Impact of a Two-Dimensional Steep Hill on Wind Turbine Noise Propagation"

_Wind Energy Science, 2024_

## Referee Comment (RC2)

Review of the paper "Impact of a Two-Dimensional Steep Hill on Wind Turbine Noise Propagation".

Submitted to **Wind Energy Science**
Article number # : WES-2024-36
July 2, 2024

**Recommendation**: Minor revision.

**Summary**:

The paper explores the complex dynamics of wind turbine noise propagation in hilly terrains using advanced numerical simulations. By solving the linearized Euler equations in a moving frame that follows the wavefront, and employing LES to simulate turbulent flow around the hill and wind turbine, the study provides a comprehensive analysis of how terrain topography affects sound pressure levels downwind. The topic is of significant interest, the methodology is robust, and the findings contribute substantially to our understanding of terrain - turbine noisy propagation dynamics. However, several issues must be addressed before the manuscript can be recommended for publication. My comments are categorized as either 'Major concerns' or 'Minor concerns', with the former focusing on conceptual technical critiques, and the latter highlighting grammatical and spelling errors.

**Major concerns**:

- **(1)**: In section 2.2, it mentions that Propagation model is solved in a 2D domain i.e., $\boldsymbol{v} = (u, w)$. The LES solver provides 3D velocity components, including the component $(v)$. However, the propagation model operates in a 2D domain, which might not directly account for this additional dimension. Clarification on how the extra velocity component is integrated or approximated in the 2D propagation model is needed.

- **(2)**: In Figs 15-18, the values near the turbine are not available, please explain why this happens? and how do the authors pick this range of this?

**Minor concerns**:

- **(1)**: In page 5, the manuscript mentions that " For each configuration, LES are performed for a truly neutral ABL, with and without the turbine inside the flow", please clearly define what's truly neutral ABL.

**References**

---

## Author Comment (AC1)

**Response to Referee 1**

**Referee 1:** In the manuscript, the authors conduct an extensive systematic study of wind turbine noise in various terrain scenarios. The computational approach combines the mean velocity from LES with sound propagation using linearized Euler equations. The authors consider no terrain, a wind turbine upwind of a hill, and a wind turbine atop a hill. The analysis is extensive, the conclusions are consistent with prior work, and the authors are careful to make comparisons to similar work in the past. Overall, the authors demonstrate the utility of the linear Euler equations approach. However, there are a few issues that need to be addressed.

**Response:** We would like to express our sincere gratitude for your dedicated time and effort in reviewing our manuscript. We are particularly grateful for your positive comments on our extensive systematic study, computational approach, and efforts to compare our findings with prior work. Your recognition of the utility of the linearized Euler equations approach is also greatly appreciated. Below, we provide a detailed response to each point, addressing your questions and recommendations.

**Specific Comments**

**Referee 1:** The authors use the mean velocity field from LES but do conduct analysis that is inherently unsteady with the single blade sound. What is the influence of using the mean velocity on the single-blade analysis?

**Response:** Here, the unsteady character of our analysis arises because the blade of the turbine does not perceive the same flow according to its position. This leads to modification of the sound emitted by the blade but also to modification of the focusing pattern because of the blade position inside the wake. Including the unsteady component of the flow would, therefore, likely result in less pronounced sound focusing, as variations in wind speed would smooth out the focusing zones. Furthermore, the signals would not be perfectly periodic as variations would arise due to the unsteady flow. This could result in additional amplitude modulation. This would also result in a more complex post-processing and analysis.

Assessing the impact of the unsteady flow features on sound propagation is not yet computationally feasible within the current framework. This would require a fully integrated calculation of sound propagation with a full three-dimensional time-dependent velocity field in the large eddy simulations.

We now address these considerations in the Model assumptions section, see lines 77-82.

Furthermore, what is the influence of using the mean velocity altogether? Could there be any additional focusing/defocusing of sound due to the unsteady flow field?

**Response:** In general, adding the unsteady component of the flow will induce diffraction of the sound wave. This leads to less efficient focusing but can increase the energy inside the shadow regions.

**Referee 1:** To assess the influence of the wind speed, which the authors deem to be less important, the LES results are simply scaled. How accurate is this scaling on even the mean velocity profile? For such a strong conclusion, the mean velocity scaling needs to be validated.

**Response:** This study considers neutral atmospheric boundary layers, where the wind speed scales with the friction velocity $u_*$. This scaling is well-accepted within the framework of our model. However, it is important to acknowledge that our model does not account for more detailed atmospheric effects, such as atmospheric stability, which can significantly impact flow profiles and, therefore, atmospheric sound propagation.

Increasing wind speed raises overall sound pressure levels and shifts the locations of focusing zones. Despite these changes, the main physical effects, such as sound focusing due to the wind turbine and hill wake, remain qualitatively the same. Therefore, we chose to focus primarily on one wind speed for this study.

We now address these considerations in the Cases studied section, see lines 143-147.

**Referee 1:** Only a single hill is considered whose height is the same as the wind turbine rotor diameter and hub height. What is the role of the hill characteristics in the conclusions? Would a smaller/larger hill change the conclusions? Would a less steep hill change the conclusions? These would be very welcome additions to a revised manuscript.

**Response:** Our study aims to demonstrate that the developed simulation framework can effectively simulate sound propagation in complex terrain and to study the qualitative effects like sound focusing due to the hill and the turbine wake. Our key finding is that terrain topography significantly affects sound propagation, including via its influence on the flow and wind turbine wake. Additionally, the placement of the turbine relative to the hill is crucially important. We believe these general conclusions hold, and our method provides a robust way to account for these effects. We modified the introduction to make these objectives more explicit, see line 60.

We agree that the hill characteristics, such as height and steepness, will impact sound propagation. A smaller or less steep hill would likely reduce terrain-induced propagation effects due to weaker flow disturbances, while a larger or steeper hill could enhance these discussed effects. While considering that a variety of hill shapes is beyond the scope of this study, we discuss these limitations at the end of the manuscript to make the reader aware of their potential impact, see lines 451-453.

We would like to thank the referee again for the time and effort spent and the constructive comments that have helped us improve the manuscript.

**Response to Referee 2**

**Referee 2:** The paper explores the complex dynamics of wind turbine noise propagation in hilly terrains using advanced numerical simulations. By solving the linearized Euler equations in a moving frame that follows the wavefront, and employing LES to simulate turbulent flow around the hill and wind turbine, the study provides a comprehensive analysis of how terrain topography affects sound pressure levels downwind. The topic is of significant interest, the methodology is robust, and the findings contribute substantially to our understanding of terrain - turbine noisy propagation dynamics. However, several issues must be addressed before the manuscript can be recommended for publication. My comments are categorized as either 'Major concerns' or 'Minor concerns', with the former focusing on conceptual technical critiques, and the latter highlighting grammatical and spelling errors.

**Response:** We would like to express our sincere gratitude for your dedicated time and effort in reviewing our manuscript. We are particularly grateful for your positive comments on the importance of the study and for stating that it significantly enhances our understanding of the interaction between terrain and turbine noise propagation.

Below, we provide a detailed response to each point, addressing your questions and recommendations.

**Referee 2:** Major concerns:
**Referee:** (1): In section 2.2, it mentions that Propagation model is solved in a 2D domain i.e., $v = (u, w)$. The LES solver provides 3D velocity components, including the component (v). However, the propagation model operates in a 2D domain, which might not directly account for this additional dimension. Clarification on how the extra velocity component is integrated or approximated in the 2D propagation model is needed.
**Response:** Using two-dimensional models to calculate sound propagation is common in the literature (see [1, 2]). In our case, the mean velocity field used in the 2D propagation model results from the projection of the 3D LES velocity fields in the direction of propagation. We added a comment on this in the manuscript, see line 115.

The reason for using 2D instead of 3D is the large range of frequencies and propagation distances that need to be considered, which become extremely computationally intensive in 3D. It should be noted that 2D simulations neglect horizontal refraction and 3D simulations are required to fully describe the effect of the wake on wind turbine noise propagation [3]. We now explicitly address these limitations in the Model section, see lines 109-111.

In ongoing work, we address this aspect, which goes beyond the present study. We find that horizontal refraction has a limited effect on the overall sound levels for neutral conditions, which is our focus here. For stable conditions, where velocity gradients are sharper, these effects can be stronger.

**Referee 2:** (2): In Figs 15-18, the values near the turbine are not available, please explain why this happens? and how do the authors pick this range of this?
**Response:** The reason for this is that the source model for the wind turbine noise is not valid in the immediate vicinity of the turbine. This is briefly mentioned in the wind turbine noise model section. We have now clarified in the manuscript that the grey area corresponds to this zone, see lines 338-340.

**Referee:** Minor concerns:
**Referee:** (1): In page 5, the manuscript mentions that " For each configuration, LES are performed for a truly neutral ABL, with and without the turbine inside the flow", please clearly define what's truly neutral ABL.
**Response:** By 'truly neutral,' we refer to an atmospheric boundary layer that is pressure-driven and where the temperature field is not modeled. We have now further clarified this in the section Cases studied, see lines 132-133.

We thank the referee again for the constructive comments and suggestions, which have helped us further improve the manuscript.

**References**

[1] Barlas, E., Zhu, W. J., Shen, W. Z., Kelly, M., and Andersen, S. J.: Effects of wind turbine wake on atmospheric sound propagation, Appl. Acoust., 122, 51–61, https://doi.org/10.1016/j.apacoust.2017.02.010, 2017.

[2] Cotté, B.: Coupling of an aeroacoustic model and a parabolic equation code for long range wind turbine noise propagation, J. Sound Vib., 422, 343–357, https://doi.org/10.1016/j.jsv.2018.02.026, 2018.

[3] Heimann, D. and Englberger, A.: 3D-simulation of sound propagation through the wake of a wind turbine: Impact of the diurnal variability, Appl. Acoust., 141, 393–402, https://doi.org/10.1016/j.apacoust.2018.06.005, 2018.